# A Review of Potential Role of Capsule Endoscopy in the Work-Up for Chemotherapy-Induced Diarrhea

**DOI:** 10.3390/healthcare10020218

**Published:** 2022-01-24

**Authors:** Takayuki Ando, Miho Sakumura, Hiroshi Mihara, Haruka Fujinami, Ichiro Yasuda

**Affiliations:** Third Department of Internal Medicine, Faculty of Medicine, University of Toyama, Toyama 930-0194, Japan; princess_and_pea_78@yahoo.co.jp (M.S.); m164.tym@gmail.com (H.M.); haruka52@med.u-toyama.ac.jp (H.F.); yasudaich@gmail.com (I.Y.)

**Keywords:** chemotherapy-induced diarrhea, fluoropyrimidine, capsule endoscopy, small intestinal mucosal injury, gastrointestinal cancer

## Abstract

Chemotherapy-induced diarrhea (CID) is a common, severe side effect of chemotherapy, immunotherapy, and targeted therapy. Because patients are more prone to continuing chemotherapy if they do not suffer from CID, appropriate diagnosis and monitoring of this disease are essential. However, suitable monitoring methods are yet to be developed. To date, several studies have shown that small-bowel capsule endoscopy (SBCE) is useful in visualizing the entire small intestinal mucosa and detecting small intestinal abnormalities, including bleeding, malignant tumors, and mucosal injury, associated with the use of nonsteroidal anti-inflammatory drugs and low-dose aspirin. Currently, limited studies have evaluated the small intestinal mucosa using SBCE in patients receiving fluoropyrimidine-based chemotherapy or immune checkpoint inhibitors. These studies have reported that small intestinal mucosal injury is common in patients with severe fluoropyrimidine-induced diarrhea. SBCE might be a useful screening method for the early detection of enterocolitis induced by immune checkpoint inhibitors. SBCE may be a powerful tool for the diagnosis and monitoring of CID, and understanding its indication, contraindication, and capsule-retention risk for each patient is important for clinicians.

## 1. Introduction

Most patients with cancer receive curative or palliative chemotherapeutic intervention throughout their treatment course. Gastrointestinal toxicities, including nausea, vomiting, ulceration, bleeding, constipation, and diarrhea, are often the major causes of treatment delays, dose adjustment, and treatment discontinuation during chemotherapy. Chemotherapy-induced diarrhea (CID) is one of the most common side effects of cancer therapy. Severe diarrhea is a life-threatening condition associated with dehydration and sepsis [1]. The drugs that are most frequently associated with diarrhea are 5-fluorouracil (5-FU) and irinotecan; the mechanisms underlying the action of these drugs are the inhibition of thymidylate synthase and topoisomerase I, respectively [2]. In addition, epidermal growth factor (EGF) receptors (EGFRs) and EGFR tyrosine kinase inhibitors (TKIs) are known to induce excess chloride secretion, which, in turn, causes secretory diarrhea [3,4]. Moreover, immune checkpoint inhibitors (ICIs) can induce colitis-associated diarrhea, which is endoscopically similar to ulcerative colitis [5,6]. Therefore, appropriate diagnosis and monitoring of diarrhea during chemotherapy are important, considering the fact that the absence of diarrhea would allow patients to continue chemotherapy, which would lead to a better disease prognosis. However, small intestinal mucosal findings have not been clarified in patients with CID because appropriate surveillance methods have not been developed yet.

Recently, small-bowel capsule endoscopy (SBCE) has been reported to play a pivotal role in the diagnosis of small-bowel disorders and, thus, has been widely used because of its noninvasive nature [7]. To date, several studies have shown that SBCE can be used to visualize the entire small intestinal mucosa and detect abnormalities, including bleeding, malignant tumors, and mucosal injury, associated with the use of nonsteroidal anti-inflammatory drugs (NSAIDs) and low-dose aspirin [8,9,10]. These studies have shown a possible use of SBCE for the evaluation of small-intestinal abnormalities in patients with CID. Therefore, this review sought to summarize the current state of literature on CID, with a focus on gastrointestinal cancer and the future possibility of using SBCE for the diagnosis and monitoring of CID.

## 2. Initial Assessment of CID

When diarrhea occurs during chemotherapy, its severity is primarily evaluated according to the Common Terminology Criteria for Adverse Events (CTCAE) (Table 1) [11].

From the perspective of CID management, the patient’s general condition should be classified as either “complicated” or “uncomplicated” because this categorization can help determine the most appropriate course of action [12,13]. Complicated diarrhea is defined according to the American Society of Clinical Oncology guidelines, which is as follows: CTCAE grade 3 or 4 diarrhea or grade 1 or 2 diarrhea with one or more additional signs or symptoms, including cramping, nausea/vomiting (grade ≥ 2), decreased performance status, fever, sepsis, neutropenia, frank bleeding, and dehydration. Uncomplicated diarrhea is defined as grade 1 or 2 diarrhea with no complicating symptoms. Intravenous fluid and antibiotics should be administered until all symptoms have resolved. Moreover, clinicians perform stool cultures (for *Clostridium difficile*, *Escherichia coli*, and other infectious organisms that may cause colitis), complete blood count, electrolyte panel test, and computed tomography (CT) to exclude infectious diarrhea and neutropenic enterocolitis [14]. Neutropenic enterocolitis is one of the most crucial differential diagnoses; this condition is associated with a neutrophil count of <500/L, fever, abdominal pain, and bowel-wall thickening [15]. The primary elements of disease onset appear to be intestinal mucosal injury combined with neutropenia and the immunocompromised state of the affected patients. These initial conditions often lead to intestinal edema, engorged vessels, and a disrupted mucosal surface, which make patients more vulnerable to bacterial intramural invasion. Therefore, its pathogenesis appears to overlap with that of CID, although the pathogenic mechanism is yet to be completely understood.

## 3. Agents Associated with Diarrhea in Gastrointestinal Cancer

Several clinical trials for each site of malignancy have clarified the frequency of diarrhea in patients receiving systemic chemotherapy. The proportions of patients suffering from diarrhea in recent, important clinical trials, which were based on the guidelines of National Comprehensive Cancer Network, focusing on systemic chemotherapy for gastrointestinal malignancy, are listed in Table 2. The drugs most frequently associated with diarrhea are fluoropyrimidine and irinotecan. Recently, targeted therapy and ICIs have been reported to induce severe diarrhea.

### 3.1. Fluoropyrimidine

The thymidylate synthetase inhibitor 5-FU interrupts DNA synthesis, which then leads to cell death by apoptosis. 5-FU is the primary agent used in systemic chemotherapy, particularly for gastrointestinal cancer, and its prodrugs such as capecitabine, S-1, and oral tegafur/uracil are known to induce similar effects and exhibit a similar toxicity profile [66,67,68].

The risk of diarrhea increases with the addition of leucovorin [69]. This is particularly evident in a combination therapy comprising intravenous 5-FU and irinotecan as both of these drugs are known to exert direct toxic effects on the intestinal mucosa. In trials with weekly bolus containing 5-FU and leucovorin for colorectal cancer, 15% patients suffered from grade 3 or 4 diarrhea [70]. Similarly, oxaliplatin combined with intravenous 5-FU has shown increased rates of gastrointestinal toxicity [71]. In addition, genetics might also contribute to drug-specific toxic effects as a previous study demonstrated that dihydropyrimidine dehydrogenase (*DPYD*) deficiency was associated with reduced clearance of and prolonged exposure to fluoropyrimidines. The most common genetic mutation observed in *DPYD* is an exon 14 skip mutation, which is a G-to-A change in the 5′ splicing recognition site of intron 14; this mutation is observed in 1–2% of the population [72]. Homozygous mutations in *DPYD* are considered scarce and occur in 1 per 5000–10,000 patients; however, these mutations are associated with rapid and severe myelosuppression, skin toxicity, mucositis, and diarrhea [72].

The pathophysiology of fluoropyrimidine-induced diarrhea is not fully understood, although earlier studies have reported that 5-FU induces the loss of crypt and villous cellularity through apoptosis and the inhibition of cell cycle progression [73]. The involvement of inflammatory mediators in the pathogenesis of intestinal mucositis and diarrhea was investigated in several studies [74]. Tumor necrosis factor-alpha and interleukin-1beta expression has been found to be highly elevated in 5-FU-treated mice [75]. In addition, NF-κB, which is a central coordinator of the innate and adaptive immune responses, is activated in the small intestinal mucosa 2 days after 5-FU administration [76]. Other studies have also demonstrated the involvement of nicotinamide-adenine-dinucleotide-phosphate-oxidase-dependent reactive oxygen species generation in phagocytes. In a previous study, CT revealed the increased wall thickness of the small intestine, and ileal biopsy with colonoscopy in patients with 5-FU-induced diarrhea revealed markedly acute and chronic inflammation [77].

### 3.2. Irinotecan

Irinotecan is a semisynthetic prodrug analog of camptothecin, and its anticancer effect is based on the inhibition of nuclear topoisomerase I [78]. Irinotecan is enzymatically converted by de-esterification into its active metabolite SN-38, which forms a cleavable complex by binding to topoisomerase I [79]. Currently, irinotecan is widely used in combination other regimens for the treatment of advanced colorectal and pancreatic cancers.

Irinotecan induces early- or delayed-onset diarrhea, which is defined as diarrhea that occurs >24 h after irinotecan administration. These phases result from different pathological mechanisms. Early-onset diarrhea is caused by the acute cholinergic properties of irinotecan and is often accompanied by other symptoms of cholinergic excess, such as abdominal cramping, rhinitis, lacrimation, and salivation. This toxicity is easily controlled with atropine. Conversely, SN-38 induces irreversible DNA damage in cancer cells, and its accumulation in the intestinal mucosa is believed to be responsible for enterotoxicity in delayed-onset diarrhea. The risk of grade 3 or 4 diarrhea during irinotecan treatment is increased in patients with Gilbert’s syndrome, which is characterized by decreased bilirubin glucuronidation. Therefore, homozygosity for UGT1A1 * 28 and * 6 alleles leads to decreased UGT1A1 expression or activation, increased myelosuppression, and severe diarrhea risks [80,81,82].

The pathophysiology of delayed-onset diarrhea is associated with the following three mechanisms: (1) direct damage to the intestinal epithelium, (2) infiltration of inflammatory cells that release immunogenic mediators, and (3) bacterial dysbiosis. Studies using rodent models have suggested that SN-38 is transformed into SN-38 glucuronide (SN-38G) by glucuronyltransferase in the liver; the latter is a much less toxic metabolite that is excreted into the gastrointestinal tract via bile. In stool, however, SN-38G can be hydrolyzed by β-glucuronidases of the gastrointestinal bacteria, and it then reverts into the SN-38 form and causes damage to the mucosa during drug excretion; however, this mechanism has not been confirmed in humans [83,84].

### 3.3. Targeted Therapy

The EGF and vascular endothelial growth factor (VEGF) pathways are crucial for tumor cell proliferation, angiogenesis invasion, and metastasis [85,86]. Therefore, the inhibition of these pathways with anti-EGFR antibodies, EGFR TKIs, VEGF antibodies, and multikinase inhibitors such as those against VEGF receptors (VEGFRs) is an effective strategy that targets the molecular basis of gastrointestinal malignancies [87,88]. In patients receiving TKIs, diarrhea might occur as early as 2–3 days after treatment administration, and its occurrence is up to 60% for all grades [89]. Grade 3 diarrhea develops in approximately 6–9% of patients, which results in dose reduction. Conversely, in patients receiving monoclonal antibodies, including cetuximab and panitumumab, grade 2 diarrhea is observed in up to 21% of patients, whereas grade 3 diarrhea is observed in approximately 1–2% [90,91,92].

The pathophysiology of targeted-therapy-associated diarrhea may be associated with excess chloride secretion because of dysregulated EGFR or VEGFR signaling, which might cause secretory diarrhea [4]. Moreover, EGFR pathway inhibition might prevent epithelial repair when combined with chemotherapy-induced local irritation due to fecal metabolites as well as transient lactose intolerance [93]. In addition, the inhibition of VEGFR signaling might lead to direct ischemic mucosal damage via the dysregulation of microcirculation in the gastrointestinal tract. Furthermore, multiple areas of colonic ulceration with perforation have been reported during treatment with multikinase inhibitors [94,95].

### 3.4. ICIs

ICIs are novel anticancer drugs whose mechanism of action depends on their interaction with the immune system. Their targets are molecules such as cytotoxic T-lymphocyte-associated protein 4 (CTLA-4), as well as programmed cell death protein 1 (PD-1) and its ligand PD-L1, which are expressed on the surface of T-lymphocytes. The adverse effects of ICIs relate to the suppression of T-cell activation. A recent systematic review and meta-analysis reported that the overall incidence rates of gastrointestinal toxicities in patients receiving anti-PD-1/PD-L1 monotherapy, ipilimumab monotherapy, and combination therapy were 1.3%, 9.1%, and 13.6%, respectively, for all grades of colitis; 0.9%, 6.8%, and 9.4%, respectively, for grade 3 or 4 colitis; and 1.2%, 7.9%, and 9.2%, respectively, for grade 3 or 4 diarrhea [96]. The predominant symptom in ICI-associated colitis is diarrhea with variable onset that depends on each treatment regimen. The median time to onset in patients receiving anti-CTLA-4 therapy is 5 weeks [97,98,99,100], whereas it is 2–4 months in patients receiving anti-PD-1 therapy [101,102]. Occasionally, ICI-associated colitis can occur even after 2 years [103].

Studies on endoscopic intervention for ICI-associated colitis have reported colonoscopic findings such as erythema, vascular pattern loss, mucosal or luminal hemorrhage, erosions, and ulcers [104]. Recently, ICI-associated colitis has been differentiated into five types based on endoscopic and histological findings; these types are as follows [104]: (1) focal active colitis, occasional foci of acute inflammation in the absence of chronic inflammation or significant crypt injury; (2) lymphocytic colitis, increase in the number of intraepithelial and lamina propria lymphocytes in the absence of crypt architectural distortion; (3) collagenous colitis, increases in the thickness of the subepithelial collagen plate and number of lymphocytes in the lamina propria in the absence of crypt architectural distortion; (4) UC-like, active chronic inflammation with goblet cell depletion and crypt architectural distortion; and (5) NSAIDs/infection-like, predominantly acute, superficial inflammation with the attenuation of crypt and/or surface epithelium.

Several guidelines have been developed to aid decision-making in the cases of patients suspected with ICI-associated colitis, and most guidelines are based on the CTCAE diarrhea grade [5,105,106,107]. All guidelines recommend the discontinuation of ICIs and early initiation of corticosteroids according to the diarrhea grade since the primary goals of these treatments are to promptly improve symptoms; avoid complications; and, wherever suitable, ultimately allow the continued use of ICIs to improve patient survival.

## 4. Endoscopic Approach for CID

To date, limited studies have examined endoscopic findings of the small intestine in patients with CID, although diarrhea or mucositis is frequently observed during chemotherapy administration. In 1999, mucosal damage of the terminal ileum diagnosed through colonoscopy was reported in six patients with colon cancer who had been receiving 5-FU-based chemotherapy [108,109]. Since then, several studies, including case reports and series, have reported similar findings in patients with fluoropyrimidine-induced diarrhea [110]. However, the extent and severity of this damage are yet to be investigated in detail given that a considerable portion of the small intestine is beyond the reach of a colonoscope. Recently, mucosal lesions in the entire small intestine have been revealed using SBCE for CID [111,112,113].

### 4.1. Current Status of SBCE and Its Possible Indication for CID

SBCE is a routine, first-line investigational tool for many small-bowel pathologies, and five platforms of SBCE have been approved and are available for use worldwide [7]. The size and weight of one of the platforms, the PillCam™ SB3 video capsule endoscope, are 26.2 × 11.2 mm and 3.00 g, respectively. Its battery ensures 11 h of work time, during which the capsule obtains 2–6 images if it is accelerated via peristalsis, after which the pictures are transmitted to the portable data recorder (Figure 1).

The primary indications for the use of SBCE include the following: (1) occult gastrointestinal bleeding, (2) suspected Crohn’s disease, (3) suspected small-bowel tumor, (4) surveillance for inherited polyposis syndromes, (5) evaluation of any abnormal small-bowel imaging, (6) evaluation of partially responsive celiac disease, and (7) evaluation of drug-induced small-bowel injury and response to medications. Specifically, SBCE can be used to monitor the deleterious effects of drugs. This modality clearly demonstrates NSAID-induced small-bowel damage such as erythema, erosions, small ulcerations, and web-like strictures [8]. Moreover, its utility has been reported during the monitoring of small intestinal mucosa in patients receiving transplants and mucosal healing of the small bowel in patients with Crohn’s disease after medical treatments [114]. However, some contraindications include (1) major abdominal surgery performed >6 months ago (relative), (2) presence of swallowing disorder, (3) noncompliance, (4) previous history/suspected small-bowel obstruction, and (5) pregnancy. Moreover, capsule retention, which is a major complication of SBCE, occurs in up to 20% of cases when it is performed in patients with suspected bowel obstruction [7]. Once retention is diagnosed, endoscopic (balloon-assisted enteroscopy) or surgical removal is deemed necessary. The use of the Agile™ Patency Capsule (Given Imaging, Yokneam, Israel) can reduce the retention rate when pretest suspected retention is high, and a conventional capsule can be safely used once the patency capsule is excreted undamaged.

Therefore, when SBCE is considered during chemotherapy, clinicians should confirm that patients satisfy the following proposed criteria: the (1) absence of any massive ascites or severe peritoneal dissemination, (2) ability to continue chemotherapy, and (3) capability of oral intake. Furthermore, depending on each patient’s condition, a patency capsule should be used to confirm intestinal patency based on the operator’s discretion.

### 4.2. Backglound Literature and SBCE Findings in Patients with CID

The following databases were searched by two authors (TA and MS): PubMed (1966–October 2021) and MEDLINE (1946–October 2021). Databases were searched using combinations of the following keywords: “diarrhea”, “capsule endoscopy”, and “cancer.”

All articles, irrespective of publication date, were considered. Studies were excluded if the (1) full text was unavailable, (2) article was not written in English, and (3) article did not mention both diarrhea and capsule endoscopy. After applying the search strategy and filtering the 44 articles obtained, four full-length articles that included discussions on capsule endoscopy in CID were identified and read by two authors. The study findings were extracted from the articles reviewed; Table 3 summarizes these findings. In these reports, primary sites of 38 patients were non-gastrointestinal, including lung, breast, ovarian, melanoma, and kidney cancers, although the sites of large population were gastrointestinal.

To date, two studies have evaluated the small intestinal mucosa using SBCE in patients receiving 5-FU-, S-1-, and capecitabine-based chemotherapy. Ota et al. performed SBCE in 16 patients with or without diarrhea and reported that the grade of diarrhea significantly correlated with the percentage of patients with a small intestinal mucosal break, which was defined as a mucosal defect (grade 0, 16.7%; grade 1, 57.1%; and grade 2, 100%; *p* = 0.016) [115]. Moreover, they quantified the number of mucosal injuries, and the number of mucosal breaks was 6.5 (range: 1–20) and 0 (range: 0–13) in the oral fluoropyrimidine and 5-FU groups, respectively. Sakumura et al. surveyed 536 patients with advanced or recurrent gastrointestinal cancer who received fluoropyrimidine-based chemotherapy [77]. Among the patients, 32 (6%) had developed complicated CID, with symptoms such as cramping, vomiting, fever, and sepsis. SBCE was performed in 13 patients with complicated CID, and small intestinal mucosal breaks developed in 8 (61.5%) patients. These findings suggested that mucosal injuries of the small intestine are common in patients with severe fluoropyrimidine-induced diarrhea and that oral fluoropyrimidine intake is an independent risk factor. Recently, to screen for CID, Shimozaki et al. prospectively evaluated the entire small intestine and colon using SBCE in 23 patients receiving ICIs [116]. Six patients (26.1%) exhibited edematous or mucosal breaks within 8 weeks after the administration of ICIs, although they did not develop colitis. This indicates that SBCE is a useful screening tool for the early detection of ICI-induced enterocolitis. No adverse events associated with SBCE, including capsule retention, were observed in any study.

## 5. Potential Utility of SBCE for CID Monitoring

Recent findings regarding SBCE do not mandate the use of this tool for the diagnosis of CID in all such patients. We believe that SBCE should be performed in patients with suspected or confirmed diagnosis of pre-existing small-bowel disease or in those who do not recover from diarrhea after stopping chemotherapy. However, the high frequency of mucosal injury in severe CID might influence the treatment strategy. For instance, clinicians may consider early administration of antibiotics for enterocolitis in febrile patients with CID, in addition to loperamide [117]. Sakumura et al. reported that of the 32 patients with complicated fluoropyrimidine-induced diarrhea, cramping, fever, and sepsis were observed in 15 (60%), 8 (32%), 6 (24%), and 3 (12%) patients, respectively [77]. This finding suggests that fluoropyrimidine-driven mucosal damage is associated with the disruption of intestinal homeostasis and induces bacterial translocation.

In ICI-associated enterocolitis, early evaluation via SBCE might be considered because it improves the clinical outcomes by identifying high-risk patients who might benefit from early administration of immunosuppressants. Patient mucosa appearing normal on endoscopic evaluation does not exclude the diagnosis of ICI-associated diarrhea, but such patients may have a superior prognosis to those with visible ulcerations [118]. Moreover, the resolution of clinical symptoms does not always reflect endoscopic remission after diarrhea treatment. Among patients receiving vedolizumab, 86% achieved clinical remission, whereas only 54% achieved endoscopic remission [119]. Furthermore, the confirmation of endoscopic remission may be important for decision-making regarding the reintroduction of ICIs because it is associated with the recurrence of enterocolitis, particularly in the nonremission state [100,101].

## 6. Future Directions and Conclusions

SBCE may be one of tools for understanding the cause of treatment-associated diarrhea and disease monitoring in patients with cancer with CID, although its indication should be limited considering its contraindication and capsule-retention risk. Further studies are warranted to confirm existing knowledge and bring us one step closer to improved clinical practices for CID.

## Figures and Tables

**Figure 1 healthcare-10-00218-f001:**
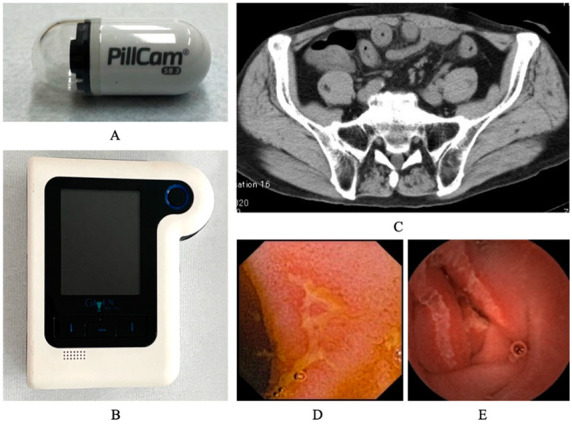
Images of capsule endoscopy of small intestinal mucosal injuries. The system of capsule endoscopy for CID contains the following elements: the capsule (**A**) and data recorder (**B**). SBCE imaging data were analyzed using reading software. CT scan suggested wall thickness of small intestine (**C**), and capsule endoscopy revealed small intestinal mucosal injuries with redness (**D**) and ulcers (**E**) in patient with CID.

**Table 1 healthcare-10-00218-t001:** Common Terminology Criteria for Adverse Events for diarrhea (version 5.0), adapted from the National Cancer Institute.

	Grade				
	1	2	3	4	5
Diarrhea	Increase of <4 stools per day over baseline; mild increase in ostomy output compared to baseline	Increase of 4–6 stools per day over baseline; moderate increase in ostomy output compared to baseline; and limiting instrumental ADL	Increase of ≥7 stools per day over baseline; hospitalization indicated; severe increase in ostomy output compared to baseline; and limiting self-care ADL	Life-threatening consequences: urgent intervention indicated	Death

ADL, activities of daily living.

**Table 2 healthcare-10-00218-t002:** Pivotal clinical trial data of frequency of diarrhea in systemic chemotherapy for gastrointestinal malignancy.

Type of Malignancy		Trial		Regimens	Proportion with Diarrhea	
		Phase	Line		Any Grade (%)	Grade 3–4 (%)
Esophagus	Sun et al. [16]	III	1st	Fluoropyrimidine, cisplatin, and pembrolizumab **	26	3
				Fluoropyrimidine, cisplatin	23	2
	Kato et al. [17]	III	Subsequent	Nivolumab **	11	1
				Paclitaxel or Docetaxel	10	1
	Kojima et al. [18]	III	Subsequent	Pembrolizumab **	5.4	0.6
				Paclitaxel, Docetaxel, or Irinotecan	20.3	3.0
Stomach	Bang et al. [19]	III	1st	Fluoropyrimidine, cisplatin, and trastuzumab *	37	9
				Fluoropyrimidine, cisplatin	28	4
	Janjigian et al. [20]	III	1st	Fluoropyrimidine, oxaliplatin, and nivolumab **	33	5
				Fluoropyrimidine, oxaliplatin	28	4
	Wilke et al. [21]	III	Subsequent	Paclitaxel, ramucirumab *	33	4
				Paclitaxel	24	2
	Hironaka et al. [22]	III	Subsequent	Irinotecan	44.5	4.5
				Paclitaxel	19.4	0.9
	Shitara et al. [23]	II	Subsequent	Trastuzumab deruxtecan	32	2
				Irinotecan or Paclitaxel	32	2
	Shitara et al. [24]	III	Subsequent	Trifluridine and tipiracil	23	3
	Kang et al. [25]	III	Subsequent	Nivolumab **	7	1
GIST	Demetri et al. [26]	III	1st	Imatinib *	44.9	2.0
	Demetri et al. [27]	III	Subsequent	Sunitinib *	29	3
	Demetri et al. [28]	III	Subsequent	Regorafenib *	45	5
	Bauer et al. [29]	III	Subsequent	Ripretinib *	28.2	NA
Neuroendocrine	Yao et al. [30]	III	1st, Subsequent	Everolimus *	34, 31	3, 7
	Raymond [31]	III	Subsequent	Sunitinib *	59	5
	Rinke [32]	III	1st	Octreotide	14.3	NA
	Caplin [33]	III	1st	Lanreotide	26	NA
Hepatocellular carcinoma	Finn [34]	III	1st	Atezolizumab **, bevacizumab *	18.8	1.8
				Sorafenib *	49.4	5.1
	Llovet [35], Cheng [36]	III	1st	Sorafenib *	39, 25.5	8, 6.0
	Kudo [37]	III	1st	Lenvatinib *	39	4
				Sorafenib *	46	4
	Bruix [38]	III	Subsequent	Regorafenib *	41	3
	Abou-Alfa [39]	III	Subsequent	Cabozantinib *	54	11
	Zhu [40]	III	Subsequent	Ramucirumab *	16	0
Biliary tract cancer	Valle [41]	III	1st	Gemcitabine, Cisplatin	NA	NA
				Gemcitabine	NA	NA
	Morizane [42]	III	1st	Gemcitabine, Cisplatin	13.5	1.2
				Gemcitabine, S-1	20.9	1.1
	Abou [43]	II	Subsequent	Pemigatinib *	37	3
Pancreatic cancer	Conroy [44]	III	1st, adjuvant	FOLFIRINOX	12.7, 84.4	NA, 19.9
				Gemcitabine	1.8, 49.0	NA, 3.7
	Von Hoff [45]	III	1st	Gemcitabine, albumin-bound paclitaxel	NA	6
				Gemcitabine	NA	1
	Moore [46]	III	1st	Gemcitabine, erlotinib *	56	6
				Gemcitabine	41	2
	Talia [47]		1st maintenance	Olaparib *	29	0
	Wang [48]	III	Subsequent	5-FU, leucovorin, liposomal irinotecan	59	13
				liposomal irinotecan	70	21
				5-FU, leucovorin	26	4
Colorectal cancer	de Gramont [49]	III	1st	FOLFOX	43.8	5.3
				5-FU, Leucovorin	58.8	11.9
	Douillard [50]	III	1st	FOLFOX, panitumumab *	NA	18.9
				FOLFOX	NA	9.1
	Heinemann [51]	III	1st	FOLFIRI, bevacizumab *	57	11
				FOLFIRI	52	13
	Cremolini [52]	III	1st	FOLFOXIRI, bevacizumab*	NA	18.8
				FOLFIIR, bevacizumab *	NA	10.6
	Cunningham [53]	III	1st	Capecitabine, bevacizumab *	40	7
				Capecitabine	35	6
	Andre [54]	III	1st	Pembrolizumab **	44	6
				Chemotherapy	62	11
	Peeters [55]	III	Subsequent	FOLFIRI, panitumumab *	NA	18.5
				FOLFIRI	NA	9.8
	Tabernero [56]	III	Subsequent	FOLFIRI, ramucirumab *	60	11
				FOLFIRI	51	9
	Van Cutsem [57]	III	Subsequent	FOLFIRI, ziv-aflibercept *	69.2	19.3
				FOLFIRI	56.5	7.8
	Overman [58]	II	Subsequent	Nivolumab **, Ipilimumab **	22	2
	Kopetz [59]	III	Subsequent	Encrafenib *, binimetinib *, and cetuximab *	62	10
				Enforafenib *, cetuximab *	33	2
				FOLFIRI/Irinotecan, Cetuximab *	48	10
	Cunningham [60]	III	Subsequent	Irinotecan, Cetuximab *	NA	21.2
				Irinotecan	NA	1.7
	Mayer [61]	III	Subsequent	Trifluridine + tipiracil	32	3
	Grothey [62]	III	Subsequent	Regorafenib *	34	7
Solid tumors						
NTRK fusion	Drion [63]	I/II	Subsequent	Larotectinib *	30	2
	Doebele [64]	I/II	Subsequent	Entrectinib *	21	1
MSI-high (noncolorectal)	Marabelle [65]	II	Subsequent	Pembrolizumab **	12	0

FOLFOXIRI, oxaliplatin, irinotecan, fluorouracil, and folinic acid (leucovorin); FOLFIRI, folinic acid (leucovorin), fluorouracil, and irinotecan; FOLFOX, oxaliplatin, fluorouracil, and folinic acid (leucovorin); NTRK, neurotrophic tyrosine receptor kinase; MSI, microsatellite instability; * targeted therapy; and ** immune checkpoint inhibitor. GIST, gastrointestinal stromal tumor.

**Table 3 healthcare-10-00218-t003:** Summary of SBCE studies in patients receiving anticancer therapy.

	Ota [115]	Sakumura [77]	Shimozaki [116]	Dore [113]
Study Design	Retrospective	Retrospective	Prospective	Retrospective
Objective	Work-Up for CID	Work-Up for CID	Screening for ICI	Screening for Chemotherapy
Number of patients who underwent SBCE	16	13	23	20
Primary malignancy				
Gastrointestinal/Non-gastrointestinal	10/6	13/0	8/15	3/17
Anticancer drug				
Fluoropyrimidine: 5-FU/S-1/Capecitabine	16 (12/2/2)	13 (2/7/4)	0	11 (11/0/0)
Molecular targeted therapy	3	4	0	3
ICIs: Nivolumab/Pembrolizumab/Nivolumab + Ipilimumab	0	0	23 (20/2/1)	0
Number of patients with CID	10	13	2	0
Diarrhea grade: 0/1/2/3	6/7/3/0	0/3/4/6	0/0/1/1	N/A
Findings of SBCE				
Edema or Redness: Negative/Positive	N/A	11/2	1/22	N/A
Mucosal brake: Negative/Positive	8/8	5/8	5/23	5/15
Adverse events of SBCE	0	0	0	0

SBCE, small-bowel-capsule endoscopy; 5-FU, 5-fluorouracil; ICIs, immune checkpoint inhibitors; CID, chemotherapy-induced diarrhea; CTCAE, common terminology criteria for adverse events; and N/A, not available.

## Data Availability

Not applicable.

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
