# Peer review of "A Review of Potential Role of Capsule Endoscopy in the Work-Up for Chemotherapy-Induced Diarrhea"

_healthcare, 2022, doi:10.3390/healthcare10020218_

Round 1

Reviewer 1 Report

The authors Ando et al. are reviewing the literature on capsule endoscopy to discuss risks and chances of its usage in chemotherapy induced diarrhea. The manuscript is well written and gives a thorough review and disscussion of the current literature on the topic.

The authors may clarify their methods section on their literature selection, especially what is meant with (4) 

"the manuscript was deemed irrelevant by the authors ".

Minor criticism: both introduction and methods unnecessarily start with a "The".

Author Response

Reviewer 1

We appreciate careful reading and constructive comments provided by Reviewer 1.

  1. Authors need to clarify criteria of literature selection on methods section.

Authors’ reply: According to your suggestion, we have described detailed methods about literature selection in Table 2 and clarified selection criteria in Table 3 (Page2, l.82 – Page3.88; Page, 8, l. 261-270).

  1. Unnecessarily start with a "The" should be deleted.

Authors’ reply: We are sorry for our carelessness. we corrected all mistakes as the reviewer indicated. Furthermore, our manuscript was re checked by a native English-speaker.

Reviewer 2 Report

I have carefully read the article written by your team and I am impressed by the data in the literature presented in this article which shows a special and dedicated work.
We also noted the detailed description of the mechanisms by which chemotherapeutics cause diarrhea in patients with oncological conditions that require their administration according to therapeutic protocols.
The idea of ​​performing small bowel capsule endoscopy (SBCE) in this category of patients is interesting, but unfortunately it is hard to believe that it can be applied on a large scale, primarily because simply, worldwide, the possibilities of performing this type of investigation is quite low.
On the other hand, it is known that these chemotherapeutics cause diarrhea after administration.
It would have been more useful to perform this type of investigation before administering chemotherapy to assess how high the risk of developing diarrhea is or to predict, based on the images obtained, the severity of diarrhea that may occur during this treatment.
In my opinion, a patient who is receiving chemotherapy and has diarrhea will have this diarrhea whether he is having SBCE or not.
Also, for those with this type of treatment and severe diarrhea, discontinuation for a certain period of time is indicated, regardless of whether they perform SBCE or not.
Although the evaluation of these patients by SBCE is a fairly convenient and somewhat safe method, conducting this type of investigation only for chemotherapy-induced diarrhea in each patient is considered an unnecessary overbidding.
I believe that this type of investigation should be performed in those patients who have a suspicion or diagnosis of pre-existing small bowel disease or in those who do not recover from diarrhea after stopping chemotherapy.
In any other case, the risk / benefit ratio should be taken into account in performing these SBCE in these patients.

Author Response

Reviewer 2

We appreciate careful reading and critical comments for this study provided by Reviewer 2.

  1. SBCE should be performed in those patients who have a suspicion or diagnosis of pre-existing small bowel disease or in those who do not recover from diarrhea after stopping chemotherapy. Furthermore, the risk / benefit ratio should be considered in performing these SBCE in any other case.

Authors’ reply: We agree on your critical comments. Recent findings of SBCE for CID suggest that not all patients with CID should receive SBCE for its diagnosis. we also believe that SBCE should be performed in specific patients considering the risk / benefit ratio as you pointed out. Therefore, we emphasize this points in section of “Potential utility of SBCE for CID monitoring‘’(Page 9, l. 297-301). However, the facts of high frequency of mucosal injury in severe CID might affect treatment strategy for CID. For instance, clinician could consider early administration of antibiotics for enterocolitis in CID patients with fever.

In addition, confirmation of endoscopic remission in ICI associated enterocolitis could be important for decision of reintroduction of ICI because the retreatment associates with recurrence of enterocolitis especially in non-remission state. This idea was added (Page 9, l. 309-Page10, 319).

Reviewer 3 Report

Interesting review article given use of SBCE for CID is not widespread 

Some comments:

Grammar needs work in the entire manuscript some :eg:1.Introduction-The Most cancer patients receive curative or palliative chemotherapeutic intervention throughout the course of treatment. 2.secretive form of diarrhea 3. pivotal role in the diagnosis of small bowel disorders because of its comfortable, noninvasive nature 4. The When diarrhea occurs during chemotherapy, its severity is primarily evaluated 56 according to the Common Terminology Criteria for Adverse Events (CTCAE)

Pacemakers and defibrillators are not considered contraindications to VCE, authors should make this correction.

The section on potential utility of SBCE for CID monitoring basically talks about management of CID and should be re written.

Author Response

Reviewer 3

We appreciate careful reading and critical comments for this study provided by Reviewer 3.

  1. The author need to correct grammar in the entire manuscript.

Authors’ reply: We are sorry for our carelessness. we corrected all mistakes as the reviewer indicated (Page1, l. 25-26; Page, 1, l. 33-36; Page 2, l. 43-45; Page 2, l. 54-56). Furthermore, our manuscript was re checked by a native English-speaker.

  1. Pacemakers and defibrillators are not considered contraindications to SBCE.

Authors’ reply: We have misunderstood about indication of SBCE. We deleted description of pacemakers and defibrillators in contraindications to SBCE (Page 8, l. 246-249).

  1. The section on potential utility of SBCE for CID monitoring should be re written.

Authors’ reply: We agree on your critical comments. Recent findings of SBCE for CID don’t suggest that all patients with CID should receive SBCE for its diagnosis. However, we believe that SBCE should be performed in those patients who have a suspicion or diagnosis of pre-existing small bowel disease or in those who do not recover from diarrhea after stopping chemotherapy. In addition, confirmation of endoscopic remission in ICI associated-enterocolitis could be important for decision of reintroduction of ICI because the retreatment associates with recurrence of enterocolitis especially in non-remission state. These ideas were added in section of “Potential utility of SBCE for CID monitoring’’ (Page 9, l. 298- Page 10, l 319).

Round 2

Reviewer 2 Report

I read your corrected manuscript carefully after the first revision and found that it has some improvements. However, the proportion in this manuscript to convince me of the usefulness of SBCE for patients with CID is very low. In fact, the title of your manuscript does not state from the outset that it is in fact the usefulness of SBCE in CID only in patients with digestive neoplasms. The title suggests that the manuscript addresses the usefulness of SBCE in CID for all patients with neoplasms of any organ. Only later, the reader realizes that it is only the neoplasms located in the digestive tract, liver, pancreas and bile ducts and diarrhea caused by the chemotherapy regimens administered for these types of neoplasms.
The largest proportion in this article is attributed to the description of some studies, the description of some chemotherapeutics and the description of the mechanisms by which they induce diarrhea.
In contrast, the part that supports the performance of SBCE in CID is too small and too unconvincing to claim that SBCE is necessary for any patient with CID or that SBCE provides information in addition to what we know about the pathogenic mechanisms by which CID occurs especially since this manuscript refers to chemotherapeutic regimens administered to patients with neoplasms already located in the gastrointestinal tract. I do not think that SBCE can detect how much of the mechanism of diarrhea is due to the chemotherapy regimen and how much is due to the location of the tumor in the digestive tract. The conclusions of this manuscript are just as unconvincing, especially since you acknowledge that "academic clinical trials might be needed."
However, please note that the treatment of patients with chemotherapeutic regimens that induce diarrhea is done in many countries around the world and that the SBCE technique is not available everywhere.
I do not deny that there are cases of CID that require SBCE but from here until SBCE is performed in any patient in the idea that new mechanisms can be discovered compared to those known to be involved in producing CID or that performing SBCE in these patients would improve the frequency of occurrence. or it would stop this side effect, it's a long way off.

Author Response

Reviewer 2

We appreciate careful reading and critical comments for this study provided by Reviewer 2.

  1. In this manuscript, the reader realizes the usefulness of SBCE in CID only in patients with digestive neoplasms.

Authors’ reply: We are sorry for our unkindness. In four articles we reviewed, half of patients had non-gastrointestinal cancers including lung, breast, ovarian, melanoma, kidney cancers. However, we summarized them as “others”. Therefore, we added the information of primary sites, and corrected the title (Page1, l. 2-3; Page, 8, l. 270-273; Table 3).

  1. The conclusions of this manuscript are just as unconvincing, especially since the acknowledge that "academic clinical trials might be needed."

Authors’ reply: We agree on your critical comments and deleted the description about clinical trials in section of future directions and conclusion (Page10, l.323-327).

Reviewer 3 Report

no further comments

Author Response

Reviewer 3

We appreciate careful reading for this study provided by Reviewer 3.

We are grateful for your comments “no further comments”.
